# Sociodemographic and Regional Determinants of Dietary Patterns in Russia

**DOI:** 10.3390/ijerph17010328

**Published:** 2020-01-03

**Authors:** Sergey Maksimov, Natalia Karamnova, Svetlana Shalnova, Oksana Drapkina

**Affiliations:** Department of Epidemiology of Chronic Non-Communicable Diseases, National Medical Research Center for Preventive Medicine of the Ministry of Healthcare of the Russian Federation, 10 bld. Petroverigskiy Lane, Moscow 101990, Russia; NKaramnova@gnicpm.ru (N.K.); sshalnova@gnicpm.ru (S.S.); ODrapkina@gnicpm.ru (O.D.)

**Keywords:** dietary patterns, principal component analysis, Russian population, socio-demographic characteristics, regional determinants

## Abstract

An empirical assessment of diets using a posteriori analysis allows us to define actual dietary patterns (DPs) in the food consumption structure of a population. This study represents an a posteriori assessment of DPs for the Russian population in general as well as their dependence on socio-demographic and regional parameters. The data were obtained from 21,923 individuals aged 25–64 years old from a Russian multicenter study of “Epidemiology of Cardiovascular Diseases in the Regions of the Russian Federation” conducted in 2013–2014. Cross-sectional study subjects were interviewed face-to-face in order to obtain data on their diet. DPs were defined using principal component analysis. Four DPs were specified as “Rational”, “Salt”, “Meat”, and “Mixed”; all these variants together accounted for 55.9% of variance. Diets in gender and age groups corresponded to those for an all-Russian population; in several regions, the defined diets differed from the all-Russian ones. More favorable diet trends were observed among women, people with no family, people not working, and urban dwellers, and diet trends were more favorable with an increase in age, level of education, and material wealth. Thus, a posteriori DPs were defined for the Russian population, which were stable in sex/age groups and were mediated by the socio-demographic characteristics of the population.

## 1. Introduction

Food consumption by humans is a rather complex heterogeneous system of eating habits associated with a tendency toward the consumption of certain foods, which can be characterized by both formed dietary patterns (DPs) and the random consumption of various foods. Demographic and socio-economic characteristics of a population can be important determinants of a healthy diet. In economically developed countries, a high income and professional and educational status are usually associated with a healthier diet, which includes a high consumption of fresh fruit and vegetables, fish and seafood, cereals, etc. [1,2,3]. At the same time, the growth of income in developing countries and the reduction in the cost of foods with a high amount of animal fat, ultra-processed foods, fast food, etc. has resulted in a low-quality diet [4,5,6]. In addition, a number of studies have shown that the regional characteristics of the population within one country can contribute to differences in dietary habits [7,8,9]. In this regard, demographic, socio-economic, and regional determinants of the dietary habits of the Russian population are of interest, as they have not been previously studied.

In the framework of the Russian epidemiological study “Epidemiology of Cardiovascular Diseases in the Regions of the Russian Federation” (ESSE-RF), the frequency of consumption of certain food groups among the Russian population was shown [10,11]. However, the consumption of certain foods can be combined with absolutely different other food products. With this connection, for example, a high level of the consumption of “favorable” fruit and vegetables can be combined in one case with the high consumption of “favorable” fish and seafood, and in the other case, with “unfavorable” sweets and pastries. Obviously, the high level of consumption of fresh fruit and vegetables in these two cases can exert quite different influences on health. Synergistic and/or antagonistic interactions between nutrients imply that, in the context of DPs, focusing on separate nutrients does not take into account all interactions and can lead to incorrect conclusions [12]. For this reason, an increasing number of studies all over the world have analyzed nutrition by taking into account the whole diet (i.e., recognizing that people consume foods in different combinations) [7,13,14].

The evaluation of DPs can be carried out using two quite different approaches [15,16]. The first includes the use of a priori defined parameters designed to determine specific DPs that represent, in general, the types, quantity, and frequency of food consumption within specific DPs [17,18]. This allows us to classify individuals within the population according to a predetermined “ideal diet”, which limits their applicability to different DP groups [19]. The second approach, which is now actively implemented, involves specifying DPs based on empirical data on food consumption by the population using statistical multivariate analysis methods [20,21]. One of these methods is factor analysis, which allows the evaluation of DPs by breaking down the consumption frequency of food groups into small components. The Russian studies using DPs have been limited to small regional cohorts [22,23,24]. As such, the aim of this study was to evaluate DPs in a wider Russian study population as well as the impact of socio-demographic and regional characteristics. We hypothesize that in Russia, there could be identified unique determinants of dietary patterns that are different from other countries. In addition, we suggest that the adherence to identified dietary patterns will depend significantly on the socio-demographic and regional characteristics of the Russian population.

## 2. Materials and Methods

### 2.1. General Characteristics of the Sample

This work was carried out as part of the multicenter epidemiological study “Epidemiology of Cardiovascular Diseases in the Regions of the Russian Federation” (ESSE-RF); 13 regions of the Russian Federation (Figure 1) took part in 2013–2014, with a total number of 21,923 individuals aged 25–64 years old. The study was carried out in accordance with the standards of Good Clinical Practice and the principles of the Declaration of Helsinki. The study protocols were approved by the Ethics Committee of National Medical Research Center for Preventive Medicine (Moscow), National Medical Research Center of Cardiology (Moscow), Almazov National Medical Research Center (St. Petersburg, Russia) as well as by collaborating centers in the regions where this study was conducted. More detailed information on sample formation and on the protocol of the ESSE-RF study has already been presented elsewhere [25]. Prior to inclusion in the study, all participants signed written informed consent. The response to the assessment was about 80%.

### 2.2. Socio-Economic Characteristics

According to family status, two groups were distinguished—the “has a family” group, which included individuals living together with a partner in an official marriage or cohabitation, and the “no family” group, which included individuals who had never been married, divorced, or lived separately as well as widows/widowers.

According to the level of education, three groups were identified: individuals with primary, secondary or secondary professional education were combined in the “Secondary” group; individuals with secondary technical and incomplete higher education were in the “Advanced Secondary” group; and individuals who completed higher education were in the “Higher” group.

The study participants were divided into groups of currently employed and non-employed for various reasons (never worked, temporarily not working, on retirement benefit, or disability benefit).

The level of income was assessed by the question: “How do you evaluate the well-being of your family compared to others?” The answers “Very poor” and “Relatively poor” were classified as “Low income”, the answer “Medium (not rich, but not poor)” as “Average income”, and the answer “Relatively prosperous” and “Very prosperous” as “High income”.

According to the type of settlement, urban and rural dwellers were distinguished.

### 2.3. Evaluation of the Consumption of Food Groups

In accordance with the research protocol, data were obtained through face-to-face interviews on the consumption frequency of 13 food groups with the following gradation: “Do not consume/Rarely”, “1–2 times a month”, “1–2 times a week”, and “Daily/Almost daily”. Among the evaluated food groups, four of them fell into the category of dairy products: “milk, kefir, yogurt”, “sour cream, cream”, “cottage cheese”, and “cheese”. In the initial statistical analysis, the frequency of consumption of these products was highly correlated. In addition, the principal component analysis revealed their combination into one stable group. In order to combine these four food groups into one enlarged group of “dairy products”, principal component analysis (PCA) with specification of a latent factor and subsequent assessment of the individual disposition of each participant to this factor were carried out. In order to reach consistency of gradation for the consumption frequency of “dairy products” with other food groups, quantitative values of tendencies were converted into four conditional categories: up to −1000, “Do not use/Rarely”; from −1000 to 0, “1–2 times a month”; from 0 to 1000, “1–2 times a week”; and from 1000 and above, “Daily/Almost daily”.

Thus, the final analysis included 10 groups of food products: “meat” (beef, pork, lamb, etc.); “fish and seafood”; “poultry” (chicken, turkey, etc.); “sausage products; organ meat” (tongue, liver, heart, etc.); “pickled products”; “cereals, pasta”; “fresh fruit and vegetables”; “legumes” (beans, lentils, peas, etc.); “sweets and pastries” (sweets, jam, cookies, etc.); and “dairy products” (milk, kefir, yogurt, sour cream, cream, cottage cheese, cheese). To apply the PCA, the frequency of consumption of food groups was expressed in conditional quantitative points: “Do not consume/Rarely”, 1 point; “1–2 times a month”, 2 points; “1–2 times a week”, 3 points; and “Daily/Almost daily”, 4 points.

### 2.4. Definition of Dietary Patterns

DPs were defined by PCA. In the course of finding the optimal number of DPs, factors with eigenvalues > 1.0 were considered, followed by a gap estimate according to the Cattell scree plot [26]. To simplify the structure of factors and to improve their interpretability, varimax orthogonal rotation of specified factors was used [27,28]. DPs were described on the basis of products with the highest absolute loading for each factor, while foods with a positive loading were characteristic of the DP, and foods with a negative loading were negatively related to the DP. Factor loading |≥0.4| of food products was considered as making a significant contribution to the factor structure. As a rule, factor loading with a value of |≥0.2| was used; however, depending on the data obtained, higher values were also used [29,30,31,32]. DPs were named according to the variables loaded on a retained component.

An individual tendency toward specified DPs was estimated with a normal distribution, an average value of 0, and a standard deviation of 1. Participants were divided into four groups (quartiles, Q) for each DP. Then, in the course of evaluating the influence of socio-demographic and regional characteristics, a comparison was made of individuals relating to the highest Q4 quartile to all others (i.e., Q1, Q2, and Q3), as done previously in [27,28,30,32].

To assess the sustainability of specified DPs among the Russian population, PCA was performed in the general sample as well as separately for men and women and the age groups 25–34 years old, 35–44 years old, 45–54 years old, and 55–64 years old for the 13 regions under analysis (Table A1, Table A2, Table A3, Table A4, Table A5, Table A6, Table A7, Table A8, Table A9, Table A10, Table A11, Table A12, Table A13, Table A14, Table A15, Table A16, Table A17, Table A18 and Table A19). With a few exceptions, the factor solutions were similar; therefore, through a comparison with other studies [33], the final factor loading using PCA varimax rotation and a selection of four factors from the general sample were used.

### 2.5. Statistical Analysis Methods

Categorical data are presented as a percent. Bilateral associations of categorical variables were evaluated by Pearson’s Chi-squared test. Multivariate analysis of associations of socio-demographic and regional variables with DPs was performed using logistic regression adjusted for gender, age, family status, education, job, material wealth, and settlement type (urban/rural). The results are represented as odds ratios with 95% confidence intervals. During the regression analysis by region, the Primorsk Territory was used as a reference group due to the largest number of study participants being from this region.

Tucker’s congruence coefficient was used for the comparison of factor solutions [34]. Congruence was characterized as “excellent” when the smallest coefficient was >0.80; “good”, from 0.65 to 0.80; “acceptable”, from 0.50 to 0.65; and “bad”, <0.50 [33].

All statistical analyses were performed using Statistica software, version 10.0 (Statsoft Inc., Tulsa, OK, USA). The critical value for statistical significance was 0.05.

## 3. Results

### 3.1. Baseline Characteristics of the Study Participants

In total, 2403 participants (11.0%) failed to fill in all the data required. Analysis of missing data revealed that 9.8% of women and 12.3% of men (*p* < 0.001) did not fill in the data. In all age groups, the proportion of individuals with missing data was as follows: 25–34 years, 9.1%; 35–44 years, 10.7%; 45–54 years, 11.1%; 55–64 years, 11.6% *p* < 0.001. In different regions, the range of missing data was usually around 6–9%, with the exception of the Samara region (3.6%), the Tyumen region (16.0%), the Volgograd region (19.3%), and the Republic of North Ossetia–Alania (25.1%), *p* < 0.001. Consequently, missing data were not random; they were biased toward the male gender, older age groups, and a number of regions. All these factors should be taken into account when evaluating the results. After the withdrawal of participants with missing data from the analysis, the final sample size was 19,520 individuals.

General characteristics of the sample are shown in Table 1. Among the study participants, there were more women (62.4%), older people (the proportion of people aged 45–54 and 55–64 was 58.7%), individuals with a family (64.6%), individuals with higher education (43.4%), employed (75.9%), with an average income (78.3%), and were urban dwellers (81.0%). The distribution of study participants by region was fairly uniform, from 6% to 10%.

### 3.2. Dietary Patterns

The variables used to carry out PCA for the whole sample and their relationships with each DP are shown in Table 2. Four DPs were specified as “Rational”, “Salt”, “Meat”, and “Mixed”, which accounted for 15.9%, 13.5%, 14.3%, and 12.2% of the total variance, respectively (i.e., 55.9% in total). The “Rational” DP was characterized by a high consumption of cereals and pasta, fruit and vegetables, sweets and pastries, and dairy products. The “Salt” DP included high-salt products such as sausages and pickled foods. The “Meat” DP included meat, fish and seafood, and poultry. The “Mixed” DP was characterized by a high consumption of fish and seafood, pickled products, and legumes.

Factor solutions for the gender, age, and region groups are presented in Table A1, Table A2, Table A3, Table A4, Table A5, Table A6, Table A7, Table A8, Table A9, Table A10, Table A11, Table A12, Table A13, Table A14, Table A15, Table A16, Table A17, Table A18 and Table A19. Table 3 shows the congruence coefficients of factor solutions by gender, age, and region in comparison with the factor solution for the general sample. In the gender and age groups, the structure of food consumption in general corresponded to the all-Russian sample, and congruence coefficients ranged from 0.97 to 0.99 (“excellent” congruence).

At the same time, specified DPs in several regions differed from the all-Russian ones. Factor solutions corresponded to all-Russian ones to the fullest extent in seven of the 13 regions: Krasnoyarsk and Primorsk Territories, Volgograd, Voronezh, Samara, Tyumen Regions, and the Republic of North Ossetia–Alania. The congruence of factor solutions seemed to be “excellent” and ranged from 0.86 to 0.99. In the Orenburg region as well as in the general sample, a 4-factor solution was obtained; however, factor congruence ranged from “acceptable” to “excellent”. In another two regions, factor solutions that differed from the common sample were even more significant—there was no “Mixed” DP food consumption structure amongst the residents of St. Petersburg, and in the Ivanovo region, “Rational” DP and “Mixed” DP were combined into a single “Rational/Mixed” DP. The congruence of the three factors was “good” or “excellent”.

In the factor solutions for the other three regions, new components appeared that were not typical for the general sample. In the Vologda and Tomsk regions, “Salt” DP and “Meat” DP were combined into a single “Salt/Meat” DP. Moreover, there was no “Mixed” DP in the food structure of these regions; however, a new separate DP appeared, which was characterized by the high consumption of fish and seafood, fruit and vegetables, legumes, and dairy products, and low consumption of sausage products (only in the Vologda region). The congruency of this new DP was most probably similar to the “Mixed” DP in the general sample. In the Kemerovo region, there was no “Salt” DP, but a new separate DP appeared, which included the high consumption of fish and seafood and low consumption of sausage products and sweets and pastries. This new DP is characterized by “poor” congruency with all four DPs of the general sample.

### 3.3. Associations of Socioeconomic and Demographic Factors with DPs

Bivariate associations of socioeconomic, demographic, and regional variables with the DPs are presented in Table 4. The proportion of individuals from Q4 of the “Rational” DP was statistically significantly higher among women who were older (45–54 years old and 55–64 years old), with no family, higher education, an average income (compared to low), and living in urban environments. The “Salt” DP was more characteristic of men with having a family, who were employed, had a low or average income, and lived in rural areas. In addition, the proportion of people from Q4 of this DP decreased linearly with age and with an increase in the education level.

The tendency toward the “Meat” DP was more characteristic for men with a family and with average and especially high incomes. In addition, the proportion of individuals from Q4 of this DP increased linearly with an increase in income. The proportion of people from Q4 of the “Mixed” DP was higher in the middle age (35–44 years old) and especially in the older age (45–54 years old and 55–64 years old) groups, and among people with advanced secondary education and those who were not employed.

Multivariate associations of socio-economic and demographic variables with the DPs are presented in Table 5. Correction for covariates slightly changed the associations observed. The tendency toward the “Rational” DP ceased to be associated with family status, and at the same time, a positive association with employment appeared. Associations appeared with the “Meat” DP, for example, an increased adherence among individuals aged 45–54 years old in comparison to those aged 35–44 years old, individuals aged 55–64 years old in comparison to those aged 25–34 years old and 35–44 years old as well as decreased adherence among people with higher education in comparison to those with secondary education. The tendency toward the “Mixed” DP ceased to be associated with people with a higher education in comparison to those with an advanced secondary education, and at the same time, positive associations appeared for people with average and high incomes in comparison to those with a low income.

### 3.4. Associations of Regional Factors with DPs

The regions under study showed numerous bivariate associations with the adherence to all four DPs (Table 4). Multivariate associations by region are presented in Table 6. Adherence to the “Rational” DP in Primorsk Territory (reference group) was one of the lowest, and therefore in eight other regions, a positive association was found, with a negative association in only two regions. The opposite situation was observed with “Mixed” DP; the adherence to this DP was one of the highest in the Primorsk Territory, so a negative association was revealed in eight other regions. In comparison to the Primorsk Territory, a higher adherence to the “Salt” DP was found in four regions and a lower one in two regions. A positive association for the adherence to “Meat” DP occurred in four regions and a negative one occurred in six regions. In addition to these regional associations with the adherence to DPs in comparison to the Primorsk Territory, many other differences were found between the regions (results not shown).

## 4. Discussion

Four stable DPs were distinguished for the Russian population, conventionally designated as “Rational”, “Salt”, “Meat”, and “Mixed”. The high proportion of the explained variance of specified DPs (i.e., 55.9%) is probably associated with the relatively small number of food groups included in the PCA [21]. Data from numerous studies indicate the beneficial effect of the “Healthy” diet and, conversely, the adverse effects of “Western” or “Meat” diets on health status indicators including cardiovascular diseases [35,36], cancer [37], diabetes mellitus [38], and all-cause and CVD mortalities [39,40]. “Rational”, “Meat”, and “Mixed” DPs include both a priori healthy foods and foods that are not recommended for frequent consumption as part of a healthy diet [41].

The “Salt” DP was represented only by non-recommended products (i.e., sausage and pickled products). Specified DPs reflect the actual consumption of foods and prove that, unfortunately, there is no stable stereotype of healthy eating behavior in the Russian population similar to the Mediterranean diet in southern Europe. At the same time, factor solutions in three regions (Vologda, Tomsk, and Kemerovo regions) specify “Healthy” DPs, which can be considered as a positive change. Our study significantly contributes to the knowledge of dietary patterns of the Russian population. While earlier, in the framework of the ESSE-RF study, the frequency of consumption of certain food groups in the Russian population was shown [10,11], our article adds to this knowledge in the context of empirical approaches and specifies dietary patterns for the population of Russia.

Taking into account food products included in specified DPs, the “Rational” and “Mixed” DPs seem to be the most favorable, especially the former. The “Rational” DP includes a fairly diverse set of healthy foods (cereals and pasta, fruit and vegetables, dairy products) which are part of many DPs that are a posteriori defined as “Healthy” in many countries [33,42,43,44,45].

The “Salt” DP partially corresponds to the DPs specified in other countries and is characterized as “Unhealthy”. In other studies, the high consumption of sausage and pickled products is often included in one DP along with the high consumption of red meat [36,46,47]. The set of foods included in the “Meat” DP is of interest due to the fact that, as a rule, in a posteriori DPs of other countries, the consumption of red meat is divided from the consumption of poultry, and especially from fish and seafood [27,46,47,48,49,50,51,52,53,54]. Red meat, in this case, is included into such DPs as “Unhealthy” or “Western”, and poultry and fish into the “Healthy” DP. There are only a few available studies where these products partially or completely belong to one DP, in particular, in Polish [31,32] and Swiss [45] populations. Apparently, this DP is, for the most part, typical for the Russian population, although it is rarely observed in the DPs of other countries. It should be noted that cross-country studies of DPs have shown both country-specific DPs, and the same DPs for different countries have been identified [55].

Unfortunately, we cannot compare our results with similar Russian studies, since our study is the first concerning the analysis of empirical data on food structures in a representative sample of the Russian population. Earlier Russian studies using PCA in order to specify DPs were devoted to the study of schoolchildren [22], schoolteachers [23], and the population of one of the regions of Russia [24]. The latest of these studies [24] was carried out within the framework of the ESSE-RF and is, therefore, part of the results of our study in the Kemerovo region, with several methodological limitations. Nevertheless, it should be noted that our results are largely consistent with similar Russian studies conducted previously.

Specified DPs are quite strongly determined by the demographic and socio-economic characteristics of a population. In terms of gender, men to a greater extent prefer “Salt” and “Meat” DPs, and women “Rational” DP. Similar preferences, where women prefer healthy diets and men, in contrast, unhealthy ones often have a high consumption of red meat, fried foods, and sausage products, have been noted in many countries such as Australia [56], Mexico [46], Great Britain [47], and Switzerland [45].

With aging, the tendency toward “Rational”, “Meat”, and “Mixed” DPs increases, but that toward the “Salt” DP decreases. Other studies have tended to show an increased adherence to healthier diets with aging [45,47,57], even when studying small age ranges, as, for example, in the Australian study of the elderly population [56]. At the same time, a study conducted in the United Kingdom showed an increased tendency toward the high consumption of red meat with aging [47] which, as in our study, indicates the ambiguous features of adherence to healthy eating among older people.

Associations with education level can be characterized generally as an increase in adherence to the “Rational” DP and a decrease in the “Salt” and “Meat” DPs with increasing education level. This tendency corresponds to the literature data from other countries [45,50,56,57,58,59] and is usually explained by the lack of knowledge, culinary skills, and motivation among people with lower levels of education [2].

An increase in material wealth is generally associated with an increase in adherence in the “Rational”, “Meat”, and “Mixed” DPs and a decrease in the “Salt” DP, corresponding in whole to the literature data. On one hand, as a rule, people with high incomes adhere to healthier diets (in this study, these are the “Rational” and “Mixed” DPs) [50,57]. On the other hand, there are studies related to the number of countries (as a rule, not belonging to the group of economically developed ones) showing a high adherence to “Meat” DPs among people with a high income [27]. Of interest are the results of a Portuguese study that showed that increased material wealth after joining the European Union in the nineties led to significant changes in diet (i.e., a change from a southern European DP to a more western (meat) one [60]; that is, frequent consumption of meat products), which are often not cheap and can be considered in some countries as a marker of prosperity, which may also be true of the Russian population.

Having a family was associated with adherence to the “Salt” and “Meat” DPs, that is, to unhealthy DPs that generally do not correspond to the data from other countries [61,62]. On the other hand, a number of studies showed the ambiguity of these tendencies. For example, a Canadian study of French-speaking men in Montreal showed high adherence of those having a family to a modified Western–Salt DP, which included, among other products, large amounts of beef, pork, chicken, hot dogs or sausages, cold cuts, bacon, and grilled food [50]. These data resemble the information obtained in this study. In an Australian study of the elderly population, married women had more tendencies toward unhealthy DPs including the consumption of processed meat, while married men tended toward vegetarian DPs [56]. Thus, the influence of family status on DPs in the Russian population corresponds to the literature data from some countries.

Among the employed population, there was a tendency toward the “Salt” DP and a negative attitude toward the “Mixed” DP. Dietary patterns of the population dependent on being employed/non-employed were not considered in other studies. However, studies have shown differences in tendencies toward DPs caused by profession; people engaged in managerial or intellectual work tend toward more healthy DPs [47]. Our results demonstrated lower adherence to a healthy diet in the working group whereas some investigators have shown the opposite [63]. Of course, employees have a higher income. Although the various markers of well-being level (education, income, professional status) are correlated, they measure different phenomena and do not replace each other as indicators of a hypothetical latent social dimension [64]. In addition, numerous studies have indicated a complex relationship between employment and eating habits, associated, for example, with strategies for coping with work stress [65] as well as with heterogeneous lifestyles in strata by employment and occupational classes [66,67].

Our study revealed that the rural population tends less to the “Rational” DP, preferring mostly the “Salt” DP. Unfortunately, we could not find any other studies of empirical DPs using PCA with an analysis of the differences between urban and rural populations. At the same time, studies on the consumption of certain food groups in the number of countries showed that the rural population has more adherence to salty foods than urban residents [68]. More frequent consumption of fruit and vegetables, sweets and pastries, and dairy products (which are included in the “Rational” DP) by the urban population was also noted in many countries [69,70,71]. Moreover, according to an Indian study, the dynamics of changes in dietary pattern were clearly observed among migrants moving from the countryside to cities, with increased consumption of vegetables, fruit, pastries, and dairy products [72].

Regional differences in DPs revealed in this study indicate a significant determinism of eating habits depending on specific local living conditions and, possibly, eating traditions. Numerous studies in other countries, especially in geographically and ethnically heterogeneous ones, have also shown at times significant differences in the dietary structure of the population of different regions [33,44,73]. At the same time, the analysis of regional features in this study mostly reflects only the fact of their presence. The study of the causes of these differences and their interpretation requires fundamental analysis and reflection, possibly with the use of additional data on the socio-economic, ethnic, climatic, and other parameters of the regions under study.

### Strengths and Limitations

The strength of our results lies in the fact that they are based on a large representative sample of the Russian population, which included 13 regions of all climatic and geographical zones of the country (central part, South, the Volga Territory, Urals, Siberia, Far East), with the exception of the Far North. Our study is the first attempt to summarize the Russian characteristics of combining various food products in dietary patterns.

However, there are several limitations pertaining to the obtained results. The main limitation is the use of a short version of the nutritional questionnaire without an indication of serving size. Although rare, such options for the evaluation of DPs are included, especially in large-scale studies [32,74]. The food groups specified for this study were the foods most often consumed among the Russian population and included both healthy and unhealthy products. The lack of serving sizes made it impossible to adjust the DPs for energy consumption. However, as noted in other studies, the frequency not adjusted for energy is more sensitive to the consumption of important low-energy foods such as fruit and vegetables [16,55,56,75]; therefore, in a number of studies, the question was set if such adjustment was required [75,76].

It should be noted that the frequency of consumption of dairy products, unlike other food groups, was calculated and combined by the sum of the frequencies of consumption of four smaller groups. This, on one hand, may create some bias in the results obtained, although, on the other hand, all calculations were logically justified and verified using adequate statistical methods.

The cross-sectional nature of this study narrows the results in the context of causal evidence of the conclusions.

Another limitation is that the data on consumption frequency were collected using questionnaires and were self-evaluated parameters; however, this method of collecting information on food consumption is common practice in epidemiological studies, especially large ones.

Finally, another limitation of this study is a rather high proportion of missing data, up to 11%. In addition, these missing data are not random, which can lead to a bias in the results obtained [77,78]. However, according to some authors, the strategy for listwise deletion of missing data from the analysis, even if they are of a nonrandom nature, seems to be reasonable [79] with a sufficiently large sample size, which was carried out in the present study.

## 5. Conclusions

There are four empirical DPs in the Russian population, conventionally defined as “Rational”, “Salt”, “Meat”, and “Mixed”, which are characteristic for all gender and age groups of the population. Most of the regions under study are also characterized by these DPs, although in some regions, some deviations were observed. Each of these DPs contained both “healthy” and “unhealthy” food products, which allowed us to state the absence among the Russian population of a clearly distinguished “favorable” DP with a high enough specific weight in the food consumption structure. At the same time, in three regions, separate DPs were identified, which included the consumption of “healthy” foods and/or a negative attitude to the consumption of “unhealthy” products, which is a positive fact.

Adherence to DPs was determined by the demographic, socio-economic, and regional characteristics of the population. More positive tendencies in DPs were observed among the groups “women”, “with no family”, “who are not working”, and “live in urban environments” as well as with an increase in age, level of education, and material wealth. Revealed regional differences in adherence to DPs require further study, possibly with the use of additional data on the geographic, climatic, socio-economic and ethnic parameters of the regions.

## Figures and Tables

**Figure 1 ijerph-17-00328-f001:**
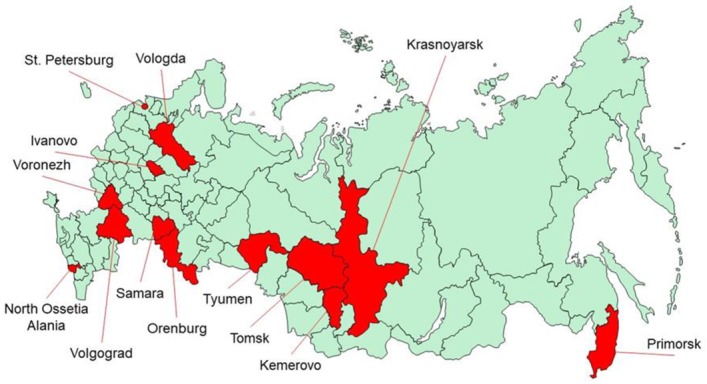
Regions of Russia participating in the study.

**Table 1 ijerph-17-00328-t001:** Baseline characteristics of the study participants.

Parameter		Amount	%
Sex	Female	12,191	62.4
	Male	7329	37.6
Age	25–34 years old	4148	21.3
	35–44 years old	3903	20.0
	45–54 years old	5432	27.8
	55–64 years old	6037	30.9
Family	No family	6905	35.4
	Has a family	12,615	64.6
Education	Secondary	5527	28.3
	Advanced Secondary	5516	28.3
	Higher	8477	43.4
Job	No	4710	24.1
	Yes	14,810	75.9
Income	Low	2098	10.7
	Average	15,291	78.3
	High	2131	10.9
Settlement type	Urban	15,817	81.0
	Rural	3703	19.0
Region	Krasnoyarsk Territory	1370	7.0
	Primorsk Territory	1903	9.8
	Volgograd region	1176	6.0
	Vologda region	1516	7.8
	Voronezh region	1480	7.6
	Ivanovo region	1731	8.9
	Kemerovo region	1469	7.5
	Samara region	1530	7.8
	St. Petersburg	1460	7.5
	Orenburg region	1445	7.4
	Tomsk region	1464	7.5
	Tyumen region	1371	7.0
	Republic of North Ossetia–Alania	1605	8.2

**Table 2 ijerph-17-00328-t002:** Factor loadings of the principal dietary patterns identified.

Product Group	Factors Identified (Dietary Patterns)			
	DP 1 Rational	DP 2 Salt	DP 3 Meat	DP 4 Mixed
Meat	−0.054	0.232	0.645 ^1^	−0.088
Fish and seafood	0.039	−0.125	0.644 ^1^	0.436 ^1^
Poultry	0.135	0.091	0.643 ^1^	−0.077
Sausage products	0.042	0.754 ^1^	0.141	−0.070
Pickled products	0.001	0.609 ^1^	0.083	0.472 ^1^
Cereals, pasta	0.471 ^1^	0.284	0.124	0.082
Fruit and vegetables	0.545 ^1^	−0.287	0.364	0.071
Legumes	0.105	0.035	−0.068	0.837 ^1^
Sweets and pastries	0.675 ^1^	0.345	−0.059	−0.186
Dairy products	0.764 ^1^	−0.229	−0.046	0.210
Proportion of explained variance, %	15.9	13.5	14.3	12.2

Note: ^1^ Factor loadings of food consumption frequency higher than 0.400 or less than −0.400.

**Table 3 ijerph-17-00328-t003:** Factors congruent in factor solutions by gender, age, and region in comparison with factor solutions in the general sample.

Group	Factor Number			
	DP 1 Rational	DP 2 Salt	DP 3 Meat	DP 4 Mixed
Female	0.99	0.99	0.99	0.99
Male	0.99	0.99	0.99	0.99
25–34 years old	0.99	0.99	0.98	0.97
35–44 years old	0.98	0.97	0.99	0.99
45–54 years old	0.99	0.99	0.99	0.98
55–64 years old	0.99	0.99	0.99	0.98
Krasnoyarsk Territory	0.99	0.94	0.98	0.97
Primorsk Territory	0.99	0.96	0.93	0.95
Volgograd region	0.91	0.86	0.96	0.97
Vologda region	0.94	0.78 ^2^	0.70 ^2^	−
New Factor ^1^	0.47	−0.40	0.58	0.62
Voronezh region	0.99	0.95	0.98	0.97
Ivanovo region	0.81 ^2^	0.82	0.72	0.53 ^2^
Kemerovo region	0.89	−	0.90	0.78
New Factor ^1^	−0.37	−0.60	0.31	0.49
Samara region	0.89	0.88	0.97	0.98
St. Petersburg	0.95	0.95	0.79	−
Orenburg region	0.80	0.70	0.98	0.64
Tomsk region	0.88	0.77 ^2^	0.65 ^2^	−
New Factor ^1^	0.52	−0.30	0.45	0.74
Tyumen region	0.91	0.94	0.88	0.95
Republic of North Ossetia–Alania	0.99	0.98	0.98	0.90

^1^ The factor solution for the new DPs was compared for congruence with all of the DPs of the factor solution of the general sample; ^2^ The factor solution of combined DPs (“Salt/Meat”, “Rational/Mixed”) was compared for congruence with the factor solutions for both DPs.

**Table 4 ijerph-17-00328-t004:** Bivariate associations of socioeconomic, demographic, and regional variables with the dietary patterns.

Parameter		DP 1 Rational		DP 2 Salt		DP 3 Meat		DP 4 Mixed	
		%	*p*-Value	%	*p*-Value	%	*p*-Value	%	*p*-Value
Sex	Female	29.2	<0.001	21.1	<0.001	23.4	<0.001	24.9	0.93
	Male	18.6		31.3		27.2		24.8	
Age	25–34 years old	22.7	<0.001	30.0	<0.001	24.4	0.12	19.9	<0.001
	35–44 years old	23.8		27.9		23.5		24.0	
	45–54 years old	27.7		24.1		25.4		26.0	
	55–64 years old	25.6		20.1		25.4		27.8	
Family	No family	26.1	0.038	23.1	<0.001	22.7	<0.001	24.8	0.94
	Has a family	24.7		25.9		25.9		24.9	
Education	Secondary	23.8	<0.001	27.9	<0.001	26.0	0.053	24.9	0.0054
	Advanced Secondary	23.7		26.2		24.3		26.3	
	Higher	27.2		22.1		24.4		23.9	
Job	No	24.4	0.14	19.7	<0.001	24.2	0.24	28.1	<0.001
	Yes	25.5		25.6		25.0		23.8	
Income	Low	22.9	0.024	24.0	0.0022	16.1	<0.001	23.1	0.13
	Average	25.6		25.4		25.4		25.1	
	High	24.8		22.1		29.4		24.9	
Type	Urban	25.6	0.0080	24.4	0.0013	24.9	0.51	24.6	0.054
	Rural	23.5		26.9		24.4		26.1	
Region	Krasnoyarsk Territory	24.2	<0.001	22.3	<0.001	30.7	<0.001	21.2	<0.001
	Primorsk Territory	19.6		23.4		26.0		32.6	
	Volgograd region	16.6		33.1		12.2		29.7	
	Vologda region	30.5		24.1		22.0		18.9	
	Voronezh region	30.1		26.2		36.6		33.4	
	Ivanovo region	27.5		31.8		18.6		14.1	
	Kemerovo region	29.7		25.0		30.0		26.1	
	Samara region	18.9		25.6		18.4		14.4	
	St. Petersburg	30.0		19.0		24.0		21.0	
	Orenburg region	32.4		27.0		30.2		32.5	
	Tomsk region	16.9		23.6		18.8		14.7	
	Tyumen region	35.9		28.2		41.4		36.6	
	Republic of North Ossetia–Alania	16.6		16.0		14.6		29.3	

**Table 5 ijerph-17-00328-t005:** Multivariate association of socioeconomic and demographic variables with the dietary patterns.

Parameter		DP 1 Rational	DP 2 Salt	DP 3 Meat	DP 4 Mixed
Sex	Male vs. Female (Ref.)	0.54; 0.51–0.59	1.57; 1.47–1.68	1.19; 1.11–1.27	1.05; 0.98–1.13
Age	35–44 vs. 25–34 (Ref.)	1.05; 0.95–1.17	0.89; 0.80–0.98	0.94; 0.85–1.05	1.29; 1.16–1.44
	45–54 vs. 25–34 (Ref.)	1.29; 1.17–1.42	0.72; 0.66–0.79	1.09; 0.98–1.19	1.44; 1.30–1.59
	55–64 vs. 25–34 (Ref.)	1.22; 1.10–1.35	0.60; 0.54–0.67	1.14; 1.03–1.26	1.47; 1.33–1.64
	45–54 vs. 35–44 (Ref.)	1.22; 1.11–1.35	0.82; 0.74–0.90	1.13; 1.03–1.25	1.12; 1.01–1.23
	55–64 vs. 35–44 (Ref.)	1.16; 1.05–1.29	0.69; 0.63–0.77	1.19; 1.08–1.32	1.15; 1.04–1.28
	55–64 vs. 45–54 (Ref.)	0.95; 0.87–1.04	0.86; 0.78–0.95	1.02; 0.93–1.12	1.04; 0.96–1.14
Family	Has a family vs. No family (Ref.)	1.02; 0.95–1.09	1.08; 1.01–1.16	1.11; 1.03–1.19	0.98; 0.92–1.05
Education	Advanced Secondary vs. Secondary (Ref.)	0.96; 0.88–1.05	0.90; 0.82–0.98	0.92; 0.84–1.01	1.11; 1.01–1.21
	Higher vs. Secondary (Ref.)	1.17; 1.08–1.27	0.66; 0.60–0.71	0.90; 0.83–0.98	1.05; 0.97–1.14
	Higher vs. Advanced Secondary (Ref.)	1.24; 1.14–1.34	0.73; 0.67–0.79	0.99; 0.91–1.07	0.93; 0.86–1.00
Job	Yes vs. No (Ref.)	1.18; 1.09–1.28	1.27; 1.16–1.38	1.01; 0.93–1.10	0.88; 0.82–0.96
Income	Average vs. Low (Ref.)	1.22; 1.09–1.36	0.95; 0.85–1.06	1.78; 1.57–2.01	1.21; 1.08–1.35
	High vs. Low (Ref.)	1.15; 0.98–1.35	0.73; 0.62–0.86	1.99; 1.69–2.35	1.30; 1.11–1.52
	High vs. Average (Ref.)	1.01; 0.90–1.12	0.77; 0.69–0.86	1.23; 1.11–1.36	1.03; 0.93–1.15
Settlement type	Rural vs. Urban (Ref.)	0.90; 0.83–0.98	1.11; 1.02–1.21	0.96; 0.88–1.05	1.08; 0.99–1.17

Note: Ref.–reference.

**Table 6 ijerph-17-00328-t006:** Multivariate associations of socioeconomic and demographic variables with dietary patterns.

Region	DP 1 Rational	DP 2 Salt	DP 3 Meat	DP 4 Mixed
Krasnoyarsk Territory	1.31; 1.10–1.56	0.91; 0.77–1.08	1.24; 1.06–1.45	0.55; 0.47–0.66
Primorsk Territory	Reference	Reference	Reference	Reference
Volgograd region	0.79; 0.64–0.96	1.56; 1.32–1.85	0.39; 0.32–0.49	0.88; 0.75–1.04
Vologda region	1.89; 1.60–2.22	0.97; 0.82–1.14	0.75; 0.64–0.89	0.50; 0.42–0.59
Voronezh region	1.77; 1.49–2.10	1.25; 1.05–1.48	1.64; 1.40–1.92	0.96; 0.82–1.12
Ivanovo region	1.50; 1.26–1.77	1.43; 1.22–1.67	0.62; 0.53–0.74	0.32; 0.27–0.39
Kemerovo region	1.77; 1.49–2.10	1.03; 0.87–1.22	1.19; 1.02–1.40	0.71; 0.60–0.83
Samara region	0.96; 0.79–1.15	1.06; 0.90–1.25	0.61; 0.51–0.72	0.35; 0.29–0.42
St. Petersburg	1.67; 1.39–2.02	0.80; 0.66–0.97	0.89; 0.75–1.07	0.52; 0.43–0.62
Orenburg region	2.05; 1.70–2.48	1.12; 0.93–1.34	1.16; 0.98–1.39	0.97; 0.82–1.15
Tomsk region	0.86; 0.71–1.04	1.01; 0.86–1.20	0.62; 0.52–0.74	0.35; 0.29–0.42
Tyumen region	2.28; 1.94–2.69	1.30; 1.10–1.53	2.06; 1.76–2.40	1.11; 0.95–1.29
Republic of North Ossetia–Alania	0.76; 0.63–0.91	0.69; 0.58–0.83	0.48; 0.40–0.58	0.81; 0.70–0.94

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
