# Peer review of "Sociodemographic and Regional Determinants of Dietary Patterns in Russia"

_ijerph, 2020, doi:10.3390/ijerph17010328_

Round 1

Reviewer 1 Report

The authors conducted an empirical assessment of diets using a posteriori analysis in actual dietary patterns in the food consumption structure of Russian population. It was found that a more favorable diet trend was seen among women, having no family, non-working, urban dwellers, as well as with an increase in age, level of education and material wealth. This is an interesting study. Appropriate methodology has been employed and the conclusions appear to be justified based on the data at hand. The authors are to be commended on describing the limitations of their study. I have a few recommendations for consideration.

Introduction. Please provide a clear hypothesis to be tested in the study. Results. The observation that most favorable diet trends was seen in people not working. This would be contrary to the general belief that people not working would have lower income and thus non healthy dietary practices.  Results/Discussion. The significance of these findings needs to be correlated to adverse health outcomes. General. Please check manuscript for typographical and grammatical errors.

Author Response

Most of the comments and suggestions of reviewers are corrected. Added Figure. Added or fixed text (highlighted in red). 10 new references were added (highlighted in green), a new numbering is highlighted in green in the text. Reviewers are given detailed comments. Please see the attachment.

Reviewer 2 Report

Language correction required;

Line 73 should be “was already presented elswere”;

Lines 76-85 – this paragraph fits better to results section;

Lines 116-120 data concerning groups of food products will be more visible if they will be placed in the table;

If it’s possible please add a map with regions used in the study to show true geographical dispersion of the regions;

Line 225- further secondary education should be advanced secondary – as previously decribed in methods and tables;

Primorsk Territory – as a apart of Kaliningrad Region or other place in Russia ?

Replace word “posterior” – it is rather used in topographic context in anatomy or histology;

Line 393 – “not working” – in line 215 was written “an average income (compared to low)” this is not consistent please explain and use proper description;

Author Response

(The authors gave the same response as above.)

Reviewer 3 Report

In this manuscript, Sergey Maksimov et al showed that posteriori DPs were defined for the Russian population, which were stable in sex/age groups and were mediated by the sociodemographic characteristics of the population.

These findings are interesting. The manuscript could be further strengthened with a few additional experiments denoted below.

1. Authors need to add the information about DPs. And, I suggest that authors need to add more reference in Introduction part.

Author Response

(The authors gave the same response as above.)

Round 2

Reviewer 1 Report

The authors have addressed all original concerns and have adequately revised their manuscript. I have no further comments.